# Simulation Analysis of Concrete Pumping Based on Smooth Particle Hydrodynamics and Discrete Elements Method Coupling

**DOI:** 10.3390/ma15124294

**Published:** 2022-06-17

**Authors:** Wang Chen, Wanrong Wu, Guoyi Lu, Guangtian Tian

**Affiliations:** 1College of Mechanical and Electrical Engineering, Central South University, Changsha 410083, China; 213701036@csu.edu.cn (W.C.); wwr@csu.edu.cn (W.W.); luguoyi@csu.edu.cn (G.L.); 2National Concrete Machinery Engineering Technology Research Center, ZoomLion, Changsha 410013, China

**Keywords:** SPH-DEM, concrete pumping, rheology behavior of fresh concrete, test validating

## Abstract

With an increase of suction efficiency of fresh concrete pumping in confined spaces, the laminar flow state will be damaged by the return flow caused by distribution value direction changes and concrete gravity. This is a fact, but one which is rarely studied. In this work, the flow state, flow velocity, and suction efficiency of fresh concrete pumping are simulated using the coupled smooth particle hydrodynamics and Discrete Elements Method (SPH-DEM). The rheological parameters and Herschel-Bulkley-Papanastasiou (HBP) rheological model are adopted to simulate fresh concrete in the numerical simulation model. The study reveals that the error between the slump experimental result and that obtained by the HBP model is negligible. A model is therefore established for numerical simulations of the suction efficiency of fresh concrete pumping. An experimental concrete pumping platform is built, and the pressure and efficiency data during pumping are collected. A comparison of the numerical simulation with experimental results shows that the error is less than 10%.

## 1. Introduction

Despite advances in computer technology and numerical simulations, several problems regarding the pumping and delivery of fresh concrete have not been solved, such as its rheological behavior, precipitation, and water secretion. Knowing the rheological behavior of concrete during pumping and delivery may help to improve performance [1,2,3]. However, traditional empirical-based simulation methods present several limitations in concrete flow predictions due to the unpredictability and complexity of the process. Therefore, numerical simulation methods are increasingly attracting attention.

Currently, there are many studies on the components of concrete and the application of various additives to improve its quality. Roussel [4] categorized the simulation methods for concrete as follows: (a) the DEM method with numerous particles; (b) the classical Computational Fluid Dynamics (CFD) method that treats concrete as a single-phase [5]; and (c) the multiphase coupled model [6].

The conventional CFD method treats concrete as a single fluid phase, ignoring many solid aggregates, steel fibers, and other components. The Eulerian method is applied to calculate concrete grid fluxes to obtain the velocity variation of the flow process and its shear deformation region [7,8,9]. Some CFD calculation software, such as OpenFoam (the OpenFoam Foundation Ltd, incorporated in England) [10], is freely available and may be used to simulate the variations in the shear deformation flow process. Choi M [11] et al. used Fluent (Ansys Ltd., Canonsburg, PA, USA) (CFD software) to analyze the rheological properties of the lubricating layer and mortar by combining experimental and numerical calculations, and found that the rheological properties of these compounds were the same. However, in their study on the rheological behavior of concrete, the pipe was generally simplified to a two-dimensional one, thereby failing to address the deeper nonlinear behavior of a non-full pipe state.

The DEM method discretizes concrete into particles such as aggregates, and slurry, and performs simulations using the friction coefficient and collision restitution coefficient among different materials. For example, Krenzer et al. [12] mixed dry and wet particles upon collision to simulate the aggregation of slurry and aggregates during concrete mixing, compared the slump experiments with the simulated analytical values, and proved the feasibility of discrete concrete as a viscoelastic material. Such expositions are unsatisfactory because they ignore the Bingham fluid flow state in pumping or piping.

The multi-phase fluid-solid coupling method is usually used to study the flow of concrete in a static pipe or an L-box without an external load. Gram A [11] utilized the CFD-DEM to measure the movement of concrete aggregate particles in pipes. The results showed that particles tend to aggregate at the bend. Similarly, Zhang Y [13] found that concrete fibers gather at the bend. CFD-DEM methods can be adversely affected by a considerable rigid body under mesh motion conditions.

The smooth particle hydrodynamics (SPH) method, in which the continuous fluid phase medium is discretized into individual particle locations, can create a realistic three-dimensional simulation. Many software packages for the SPH method are available for public use, such as the open-source Dualsphysics [14]. The SPH method is adopted to analyze the rheology of complicated materials because it can easily deal with complex media deforming equations. For example, it is widely applied in massive kinematic deformation [12,15].

The flow of fresh concrete in pipes has received the attention of many scholars, while concrete pumping has not. Most recent studies have focused on the effects of material additions on concrete properties, while the suction efficiency and concrete flow in the pumping cylinder and pipe have attracted very little attention.

In summary, previous studies on fresh concrete using DEM have paid little attention to the laminar flow state during pumping. Conventional CFD methods generally do not correspond to actual concrete composition ratios. On the other hand, most studies related to the CFD-DEM double-way coupling have focused on discontinuing the static pipes only. Therefore, the need for new research approaches to describe the continuous pumping and components of fresh concrete has been recognized.

The suction efficiency of concrete pumping is numerically simulated in this work using coupled SPH-DEM (based on the DualSPHYsics (developed by New Jersey Institute of Technology, New Jersey, NJ, USA; The University of Manchester, Manchester, UK; Universidade Vigo, Vigo, Spain; Università degli studi di Parma, Parma, Italy; Universitat Politecnica de Catalunya–BarcelonaTech, Barcelona, Spain, and Instituto Superior Tecnico, Lisbon, Portuga) open-source software coupled with the Project Chrono open-source software). Firstly, SPH-DEM is utilized to create a fresh concrete simulation and analysis model. Secondly, the practicality of the SPH-DEM to produce new concrete is proved by experimentally evaluating the numerical simulation features of the concrete using concrete aggregate grading and slump flow tests, as well as a rheometer. Then, experiments are performed using the verified completed concrete and numerical models of the pumping process. Finally, the theoretical velocity of concrete movement in the cylinder is compared to that of the numerical simulation.

## 2. SPH-DEM Theory

### 2.1. SPH Methods

SPH [16] is one of the earliest Lagrangian procedure strategies. In it, a cloud of “particles” represents the physical material of the domain. SPH control equations are calculated by convolution with a weighted kernel operating and variables of the flow field (such as velocity and pressure). Specific details of the SPH calculation solution to the N-S equation may be found in the literature, e.g., by Monaghan J J [17] et al. For viscous flow, the Navier–Stokes acceleration equation given by Ye T [18] is expressed as follows:(1)$$\frac{dv_{i}}{dt}=-\frac{1}{\rho }\frac{\partial P}{\partial x}+\frac{1}{\rho }\left[\frac{\partial }{\partial x_{i}}\left(\eta \left(\frac{\partial v_{i}}{\partial x_{k}}+\frac{\partial v_{k}}{\partial x_{i}}-\frac{2}{3}\delta_{ik}\nabla \upsilon \right)\right)+\frac{\partial }{\partial x_{i}}\left(\xi \nabla \upsilon \right)\right]$$
where η is the shear viscosity coefficient, and ξ refers to the bulk viscosity. Ye T [18] determined the viscosity for the solution and rapid convergence of the N-S equation. In SPH equations, the viscous factor, represented by $\Pi_{ab}$, is added to the pressure terms to yield.
(2)$$\frac{dv_{a}}{dt}=-\underset{b}{\sum }m_{b}\left(\frac{P_{a}}{\rho_{a}^{2}}+\frac{P_{b}}{\rho_{b}^{2}}+\Pi_{ab}\right)\nabla_{aW_{ab}}+g\;,\;\Pi_{ab}=-v\left(\frac{v_{ab}r_{ab}}{r_{ab}^{2}+\varepsilon h^{2}}\right)$$
where h is the smoothing length which defines the impact domain of the kernel. In this work, the Wendland Quintic kernel function was adopted, as shown in Figure 1:(3)$$W\left(r,h\right)=\alpha_{D}{\left(1-\frac{q}{2}\right)}^{4}\left(2q+1\right)\;\;\;,0\leq q=r/h\leq 2$$

For slightly compressible fluids (e.g., water), the SPH is usually approximated by replacing it with an artificially incompressible fluid, which is the basis of most finite difference algorithms. Another approach to SPH calculations is to approximate a weakly compressible fluid, which requires that the speed of sound is large enough for density fluctuations to be negligible. When the air pressure is negligible, the form can be described by Equation (4), according to Morrison [19]:(4)$$P=\frac{c^{2}\rho_{0}}{\gamma }\left[{\left(\frac{p}{p_{0}}\right)}^{\gamma }-1\right]$$
where $\rho_{0}$ is the reference density which usually equals the initial density of the boundary or floating, ρ is the density of SPH particle, $c=\sqrt{\gamma \beta /\rho_{0}}$ is the speed of sound at the reference density, and β is a constant to maintain the relative density fluctuation |δρ|/ρ. As suggested by Colagrossi [20], β = 100 means that the speed of sound is 10 times greater than that of the most wave, thereby maintaining the density variation at interval bounds of 1%. γ is set to 7.0 in Wang [21].

SPH adopts the smooth kernel function for calculations. If the particle boundary is within the particle computational domain of the SPH smooth kernel function, the second-order accuracy of the calculation and analysis can be guaranteed. However, when intersecting with the boundary, the smooth kernel function is truncated by the boundary, and its integral value is no longer 1. The computational error will cause a loss of the second-order accuracy of the SPH smooth kernel function [22]. To solve this problem, the dynamic boundary condition (DBC) was selected as the boundary condition in this work [23]. The physical density/pressure values are used in boundary particles, but the separation gap between bound and fluid particles is avoided. The DBC has been applied to simulate Newtonian [15] and Non-Newtonian fluids [21,24].

### 2.2. Rheology Model for Non-Newtonian Fluids

Inelastic fluid behavior is common in several non-Newtonian applications. Springless fluid behavior is expressed as the shear rate in follow equation:(5)$$\tau =\mu \dot{\gamma }$$
where τ is the shear stress, μ is the viscosity of the fluid, and $\dot{\gamma \;}$ is the rate of deformation. However, many fluids (like paint, lubricants, blood, mud, ice, etc.) do not follow that rule and present variable consistencies.

In rheology, the standard Bingham model [25] is usually expressed as a mathematical model, as in Equation (6):(6)$$\tau =\{\begin{array}{l} \tau_{y}+u\cdot \dot{\gamma },\tau \geq \tau_{y}\\ \tau_{y}\;\;\;\;\;\;\;\;\;\;\;\;\;,\tau <\tau_{y} \end{array}$$
where $\tau_{y}$ refers to material yield stress, Pa; τ is shear stress, Pa; u represents the plastic viscosity, Pa·s; and $\dot{\gamma }$ is the shear rate, 1/s. Equation (6) is the basic expression for visco-plastic fluids obtained the Herschel-Bulkley (HB) rheological model, showing that the effective viscosity exhibits a power law when the shear stress is above the yield stress. The mathematical model of HB is as shown in Equation (7):(7)$$\mu_{eff}=\frac{\tau_{y}}{\dot{\gamma }}=\frac{\tau_{y}}{2\dot{\gamma }}+K\cdot {\left(\dot{\gamma }\right)}^{n-1},K=\mu \cdot 2^{n-1}$$

In Equation (7), if n = 1, the HB model is reduced from a power-law expansion model to the conventional Bingham model. However, Equation (7) contains a downside in which $\mu_{eff}$ is infinite once and $\dot{\gamma }$ is joined to zero, which can result in the non-convergence of the calculation results [26]. Therefore, Papanastasiou proposed an enhanced model called Herschel-Bulkley-Papanastasiou (HBP). The enhanced HBP model is defined as Equation (8), based on Equations (6) and (7):(8)$$\mu_{eff}=\frac{\tau_{y}}{2\dot{\gamma }}\left(1-e^{-2m\dot{\gamma }}\right)+K\cdot {\left(\dot{\gamma }\right)}^{n-1}$$

In Equation (8), if m is massive enough, the HBP model is roughly equivalent to the Bingham model. Then, a second-order expansion of $e^{-2m\dot{\gamma }}$ using the Peano residual term Taylor-McLaughlin formula is obtained as Equation (9):(9)$$\underset{\dot{r}\to 0}{\lim}\frac{1}{2\dot{\gamma }}\left(1-e^{-2m\dot{\gamma }}\right)=m$$

There are two unique coefficients, m and n, in the HBP model compared with the commonly used Bingham model. A sensitivity analysis is performed to analyze the impacts of m and n on the HBP model. m and n are modified in the analysis, while the remaining parameters are constant (μ = 0.02 Pa·s and τ = 2.00 Pa). m mostly governs the initial rapid rise of shearing stress, while n primarily influences the linear or nonlinear behavior in the high shearing rate range, as shown in Figure 2.

### 2.3. DEM Method

As indicated in Figure 3, DEM discretizes the item into many particles and solves the forces among them via collision and deformation.

The particle force is divided into normal and the tangential forces. Both are made up of the material’s completely elastic collision force and damping dissipation force. The normal force can be calculated using Coulomb’s law of friction, as in Equations (10) and (11).
(10)$$F_{n}=F_{n}^{r}-F_{n}^{t}=k_{n}\delta^{3/2}-\gamma_{n}\delta^{1/2}\dot{\delta }$$
(11)$$k_{n}=\frac{4}{3}E^{*}\sqrt{R^{*}},\gamma_{n}=-\frac{lne_{ij}}{\sqrt{\pi^{2}+ln^{2}e_{ij}}}$$
where $e_{ij}$ is the average of the collision response coefficients of the two materials.
(12)$$R^{\;*}={\left(\frac{1}{R_{1}}+\frac{1}{R_{2}}\right)}^{-1},\;E^{\;*}={\left(\frac{1-v_{p1}^{2}}{E_{1}}+\frac{1-v_{p2}^{2}}{E_{2}}\right)}^{-1}$$
where $R_{i}$ is the particle radius, $E_{i}$ represents the modulus of elasticity of the material, and $v_{pi}$ is the Poisson’s ratio of the material. The tangential force is calculated using a similar method of normal force, as shown in Equations (13) and (14):(13)$$F_{t}=F_{t}^{r}-F_{t}^{c}=k_{t}\delta^{t}-\gamma_{t}\delta^{t}\dot{\delta }$$
(14)$$k_{t}=2/7k_{n},\gamma_{t}=2/7\gamma_{n}$$

## 3. Experiment Test

### 3.1. Slump Tests

Experiments on concrete-related ratios and the slump-related gradation related to slump are required to determine the judgement criteria of the pumping procedure. The concrete is proportioned and matched in accordance with the characteristics listed in Table 1.

The aggregate is sieved for size and density, and graded in order to refine the concrete for the pumping. The results are displayed in Figure 4.

The aggregate in Table 2 is classified into four grades, ranging from 19 to 9.5 mm in diameter, with the percentage of each grade stated in the diagram.

After being certified, the aggregates are mixed into the concrete. Then the mixed material described in Table 1 was slump tested. The slump test is depicted in Figure 5.

Slump tests were carried out for both proportions of the concrete described in Table 1. The results are shown in Table 3.

After that, the rheological properties of fresh concrete and the related rheological coefficient are measured, as shown in Figure 6 and Figure 7.

### 3.2. Concrete Pumping Test

The focus of this work is the response of the concrete suction process relative to an actual physical model during concrete pumping; hence, the pumping mechanism is shown in Figure 8a. The hydraulic schematic of the pumping system is depicted in Figure 8b. In this Figure, 1, 6, 8, and 17 are the pressure gauge; 2 and 16 is the electromagnetic control reversing device; 3, 13, 14, 20, and 21 are the hydraulic directional valve; and 5, 15, 22, and 7 are the pressure control valve. Actuator hydraulic motor 4 is the mixer paddle. Hydraulic cylinder 12 is the distribution valve swing cylinder. Hydraulic cylinders 18 and 19 are pumping cylinders. The pumping process by hydraulic cylinders 18 and 19 alternates direction with the oscillating hydraulic cylinder 12 to complete the pumping process.

Figure 9 illustrates the pumping effectiveness load in this mechanism.

As concrete is not transparent, it is difficult to observe the changes of internal shape during delivery. Improving the concrete pump inhalation efficiency means observing and studying the experimental process by using a concrete pump which is constantly inhaling and pushing out concrete. As a result, the performance of the total quantity of concrete inhaled by the pump can be characterized by using the weight of the concrete pump output. This yields large amounts of data and records the mass during the delivery. Before pumping, the concrete density is determined, and the concrete output squared is calculated using the concrete weight.

The hydraulic pump output flow rate can be determined, and the average efficiency in many pumping trials can be calculated using the ratio of the two flow rates. The experimental process should be performed ten times, with each group employing concrete pumps. The findings on efficiency and mass are displayed in Figure 10.

According to the experimental results, the mass of every ten pumping instances was 1890–2380 kg. The average weight of each pumping in the 12 groups was 2156 kg. The pumping efficiency was 82.1–92.61% using the hydraulic pump output flow rate as determined from the hydraulic system. The pumping efficiency for the 12 groups was 86.9%, on average.

## 4. Simulation Test

### 4.1. Tools

In this work, the HBP model was adopted to investigate the rheological behavior, and the DEM method was employed to solve the forces between the fluid and the solid. The HBP model is an excellent non-Newtonian fluid model for simulating fresh concrete, and it can accurately simulate the Bingham or Power-law models used to describe slurry flow.

### 4.2. Slump Simulation

To offer a thorough depiction of the pumping process, the concrete simulation analysis model, which is a comparison of the experimental data, first had to be calibrated. The parameters of the simulation model were defined based on the experimental results listed in Table 4.

The simulation of the Group 1 slumps is shown in Figure 11a–d.

In Figure 11a, the concrete slump vacant part is the location of aggregate generation for the DEM method, while the aggregates are SPH particles. The initial fresh concrete was mixed and stirred above the slump cylinder and entered the cylinder by gravity. Aggregates were generated in the proportions given in Table 1. Figure 11b shows the fresh concrete flowing into the slump cylinder; the yellow part indicates the concrete slurry while the rest is the concrete aggregates. Once the concrete has entered the slump cylinder, the vessel is quickly lifted, at which point the slump experimental process begins. The slump experimental results are shown in Figure 11c,d. Figure 11c is the cross-section in the X–X direction; at this time, the slump corresponding to the simulation analysis was about 190 mm, the remaining height was 110 mm, and the extension was 375 mm. In Figure 11d, the slump is shown as 400 mm. The simulation analysis results were compared with the experimental results, and the data results are shown in Figure 12.

The simulation results of Group 2 are shown in Figure 13a,b. The slump of the compared process was 180 mm, while that in the simulations was about 185 mm, and the extension were set to 300 mm and 325 mm, respectively.

The simulation analysis results of Group 2 were compared with the experimental data, as shown in Figure 14. The slump was 180 mm in the experiment and 185 mm in the simulation, and the expansions were 295 mm/290 mm and 320 mm/300 mm, respectively. The error between them was small, which suggests that the SPH-DEM method showed a good response and fitting to the actual physical model.

### 4.3. Concrete Pumping Simulation

The flow during concrete pumping is difficult to observe and represent, since it is in a restricted environment. Furthermore, there is no clear theory for how aggregate and slurry are separated in concrete, and this can only be determined by waiting for the former to accumulate, resulting in pipe obstruction. To describe the pumping process completely using the current computer technology, it is necessary to compare the pumping efficiency of a simulation analysis with the actual efficiency to determine whether they are compatible.

Therefore, the mass of concrete pushed via the pumping cylinder is approximated by that sucked by the pumping cylinder, i.e., the conservation of mass. It is critical to investigate the efficiency and mass sucked by the pumping cylinder. Figure 15 depicts the pumping simulation analysis model applied in this research; the trailer pump assembly (a), the hopper (b), the distribution valve (c), and the mixer paddle (d) are shown. The reciprocating motion of the pumping cylinder and the distribution valve play a supporting role to the pump to concrete.

The experimental process is complex. As such, simulation analyses were used to find a way to optimize the experimental prototype, which can be understood as approximating and matching the changing trend of the simulation analysis system and the actual experimental model. The actual experiment model contains too much information in the edges and corners, increasing the difficulty for the simulation analysis model to focus on real issues. Therefore, the edges and corners must be simplified. The redundant hole features in the pumping structure model shown in Figure 16a–d depict a basic simulation analysis model.

The Y–Y section is shown in Figure 16a. The main mechanism of the pumping model is illustrated in Figure 16b; it includes a mixer, a distribution valve, two pumping cylinders, and a pumping piston. The volume ratio of slurry and aggregate was determined using the experimental concrete ratio. Figure 16c shows the beginning generation location of concrete and aggregate. The piston movement velocity during pumping was set to 1 m/s, and the distribution valve change of direction time was defined at 0.2 s each time that the system realized a function, as indicated in Figure 17 and Figure 18.

The distribution valve is in the left position of the cylinder when the pumping model is at 0 s; after a reversal at 0.2 s, it moves to the right position. In accordance with the experimental procedure, the stirrer was operating at 20 rpm. Pumping in the experimental process was repeated 20 times per minute. The matching pumping speed was approximately 1 m/s, and the simulation analysis speed was assumed to be 1 m/s. The pumping cylinder of the simulation analysis model was 1.4 m long. When the suction finished after 1.4 s, the simulation analysis showed that the suction full tube position was 1.2 m, which corresponded to a single inhalation of 85.714%. The suction efficiency of the physical experiment was calculated to be 86.9%, as shown in Figure 18, with a 1.2% error. This demonstrates that the simulation analysis was quite close to the experimental process in terms of a practical fit.

The Bingham fluid in the pipe with the resolved motion velocity gives the flow velocity of concrete in the pipe via Equation (15):(15)$$u_{z}\left(r\right)=\{\begin{array}{c} \frac{\Delta PR^{2}}{4\mu_{p}L}{\left(1-\frac{r_{0}}{R}\right)}^{2},\;0\leq r<r_{0}\\ \frac{\Delta PR^{2}}{4\mu_{p}L}\left(1-{\left(\frac{r}{R}\right)}^{2}\right)-\frac{\tau_{0}R}{\mu_{p}}\left(1-\frac{r}{R}\right),\;\;r_{0}\leq r<R \end{array}$$
where $r_{0}$ is the yielding surface area defined by Equation (16)
(16)$$r_{0}=\frac{2\tau_{0}L}{\Delta P}$$
where ΔP/L is the pressure gradient along the pipe. The ratio of pumping pressure to pumping length is calculated in Figure 19.

To obtain the theoretical velocity in the respective region of *r*/*R*, the parameters of the moving process were introduced into the preceding equation. The velocity of the concrete radial motion in the pipe is shown, and the two analyses are compared in Figure 20.

The theoretical motion velocity of Bingham fluid in the pipe was well fitted to that in the numerical simulation.

## 5. Conclusions

This work aimed to develop a new method to determine fresh concrete behavior during pumping.

The results showed that the SPH-DEM method could be utilized to simulate fresh concrete through slump tests. The rheological parameters of fresh concrete were identified by a rheometer. The slump test error between simulation and experiment results was compared and shown to be less than 10%. Therefore, the SPH-DEM numerical simulation could potentially be used to react a real physical model.A numerical simulation model of the pumping process was established to analyze the effects of variations in the plastic viscosity of fresh concrete on the suction efficiency. In addition, the suction efficiency was studied experimentally. The numerical simulation results were compared with the experimental results, and the average error of suction efficiency was less than 5%.The gradient variation of the pressure loss along the pipe (Dp/L) was calculated and the theoretical flow rate of concrete in the pipe was analyzed. Compared with the numerical simulation, the theoretical velocity analysis showed that the results from the SPH-DEM numerical model approached those of the theoretical analysis. The issue of the pipe blocking mechanism is an intriguing one which should be further explored in future research.

## Figures and Tables

**Figure 1 materials-15-04294-f001:**
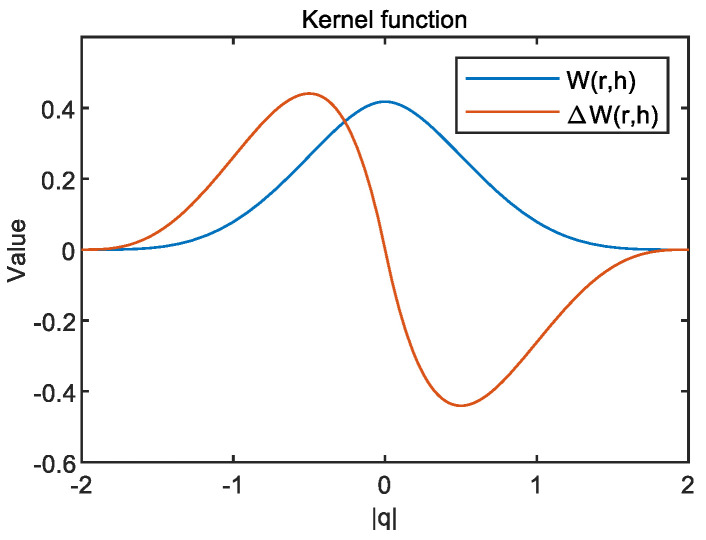
Kernel function and derivative.

**Figure 2 materials-15-04294-f002:**
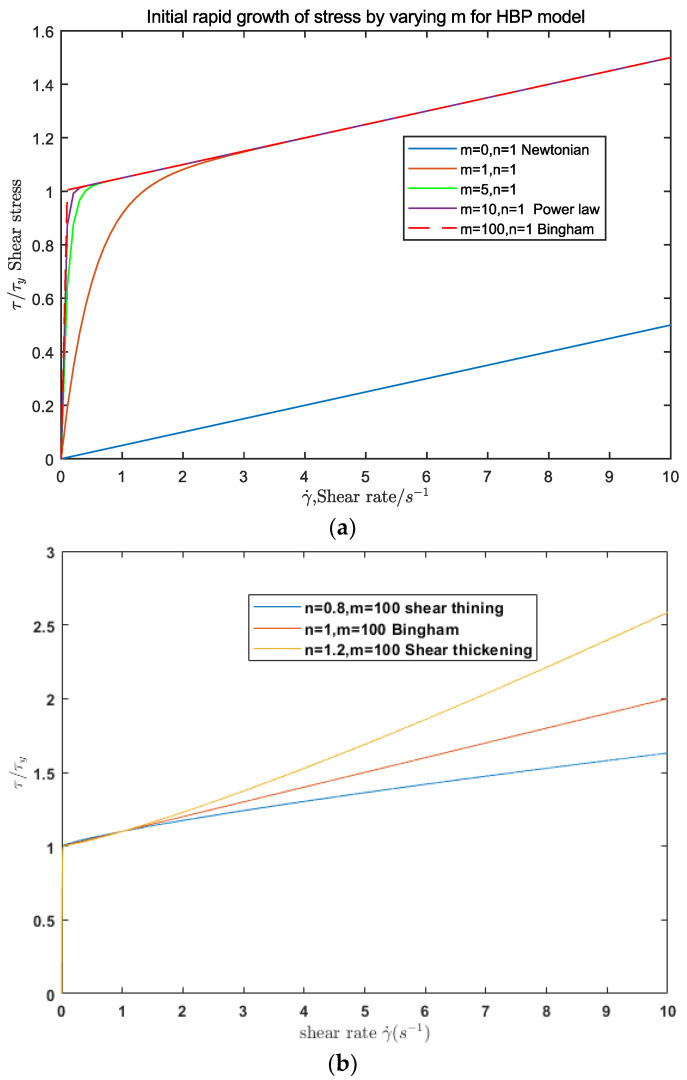
Parameter analysis of the HBP model; (**a**) is for coefficient m (*n* = 1) and (**b**) is for coefficient *n* (*m* = 100).

**Figure 3 materials-15-04294-f003:**
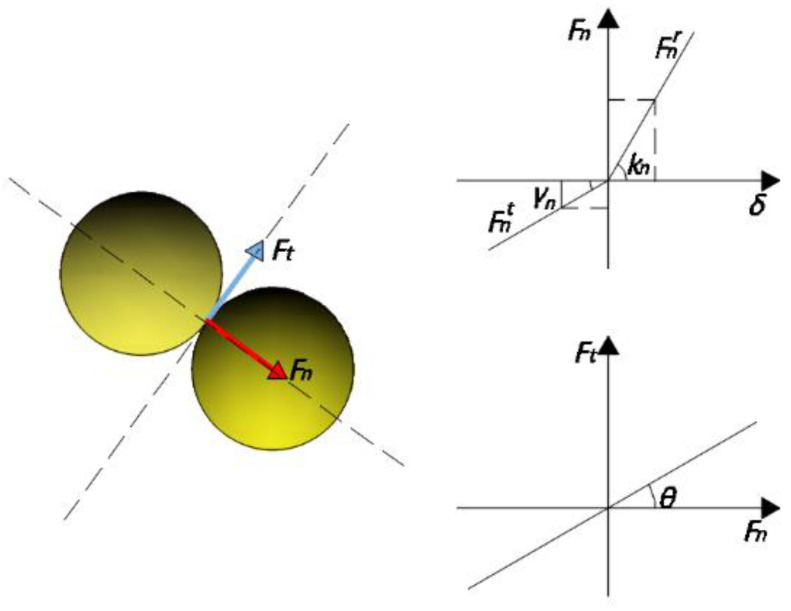
DEM calculation.

**Figure 4 materials-15-04294-f004:**
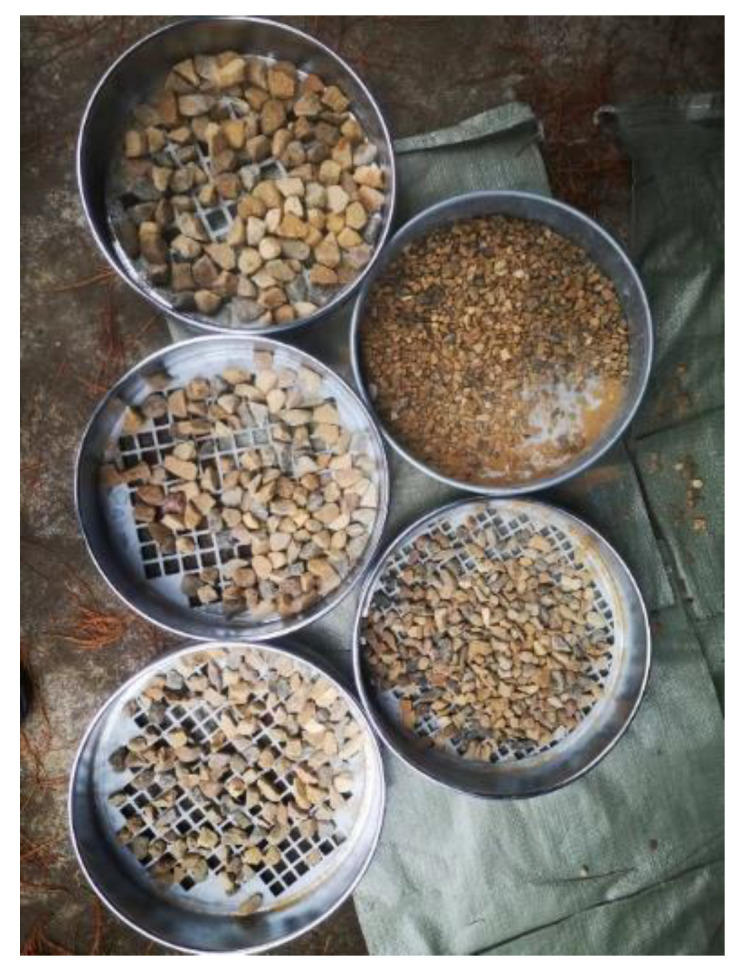
Aggregate classification results.

**Figure 5 materials-15-04294-f005:**
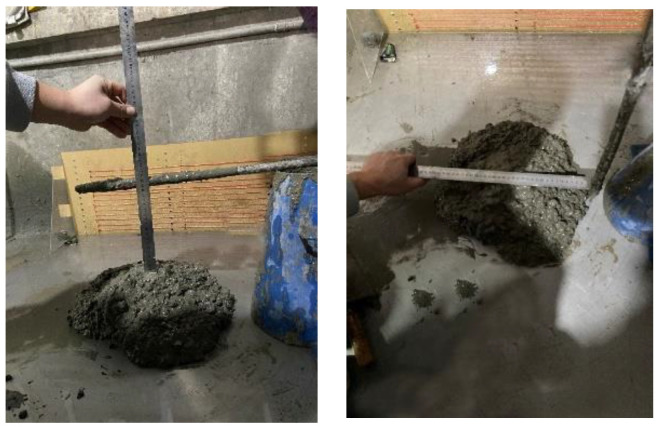
Slump tests.

**Figure 6 materials-15-04294-f006:**
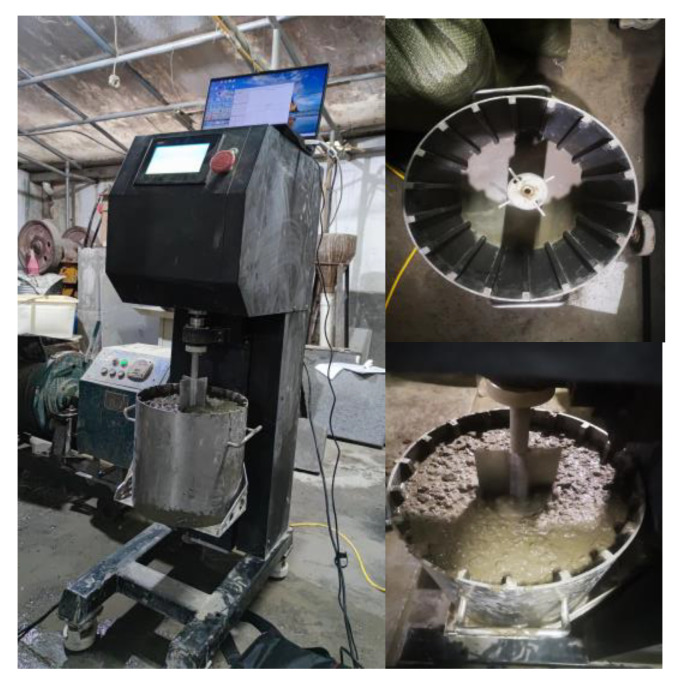
Rheometer measurement.

**Figure 7 materials-15-04294-f007:**
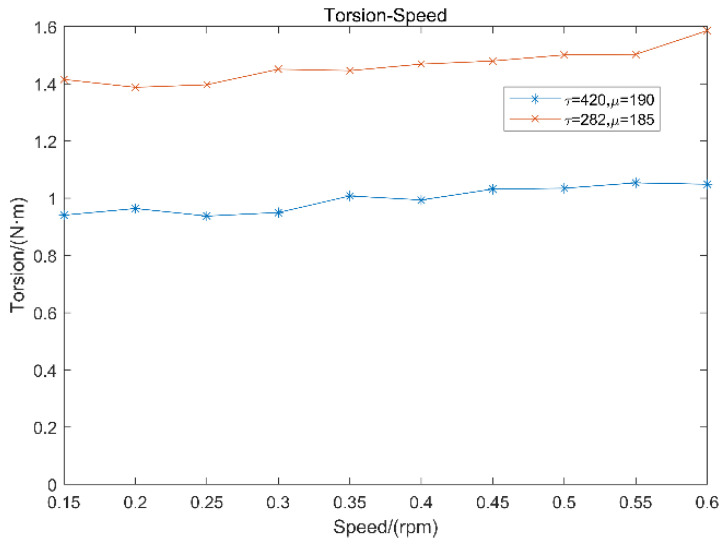
Experimental rheometer data.

**Figure 8 materials-15-04294-f008:**
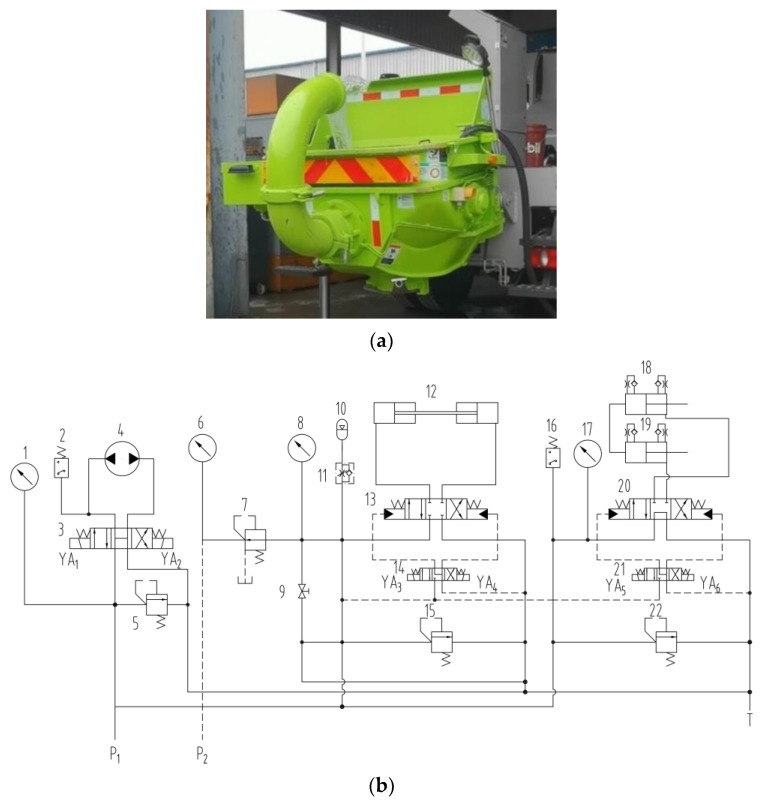
(**a**) Concrete pumping truck; (**b**) Schematics of concrete pumping system.

**Figure 9 materials-15-04294-f009:**
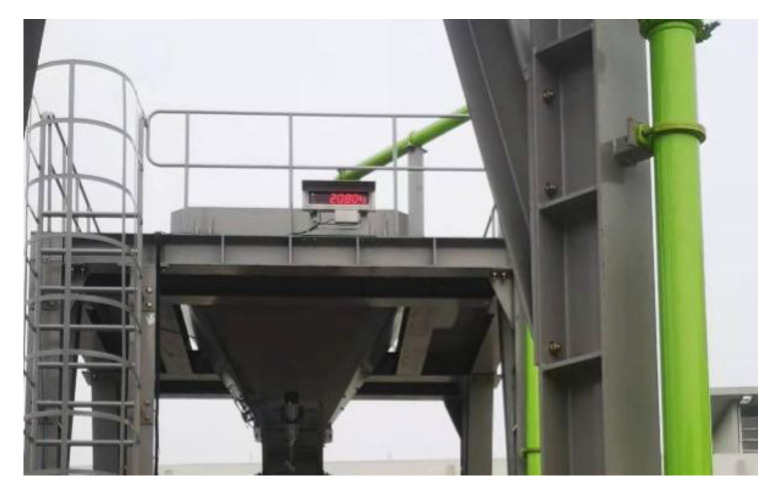
Concrete pumping test.

**Figure 10 materials-15-04294-f010:**
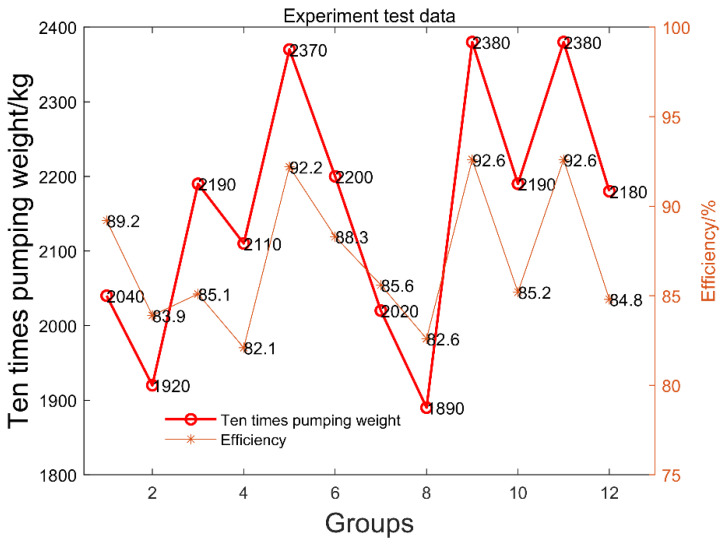
Concrete pumping efficiency and mass.

**Figure 11 materials-15-04294-f011:**
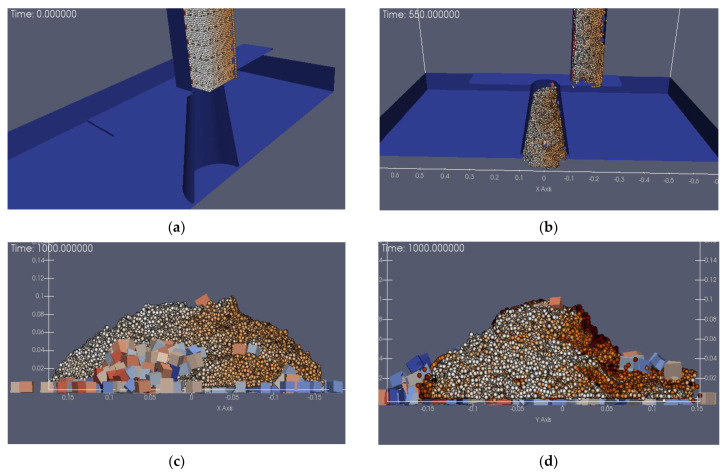
The slump numerical simulation of Group 1. (**a**) Initial state (**b**) lifting time (**c**) X–X Cross-sectional view (**d**) Y–Y Cross-sectional view.

**Figure 12 materials-15-04294-f012:**
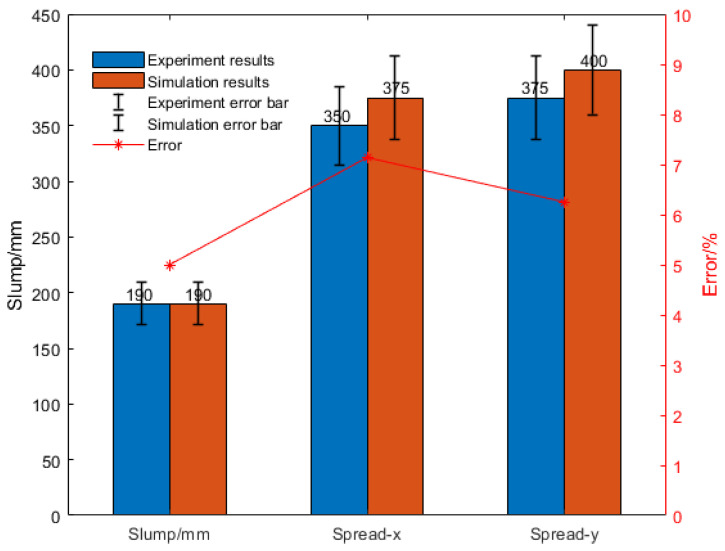
Error analysis of Group 1.

**Figure 13 materials-15-04294-f013:**
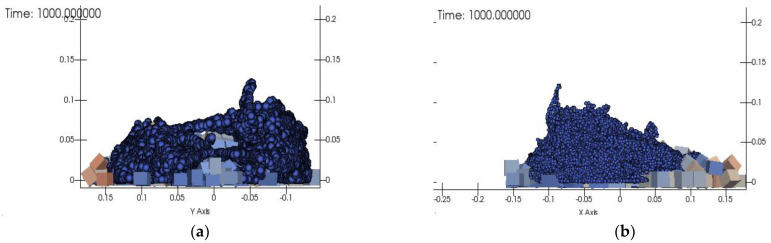
Simulation results of Group 2. (**a**) X–X cross-section view (**b**) Y–Y cross section view.

**Figure 14 materials-15-04294-f014:**
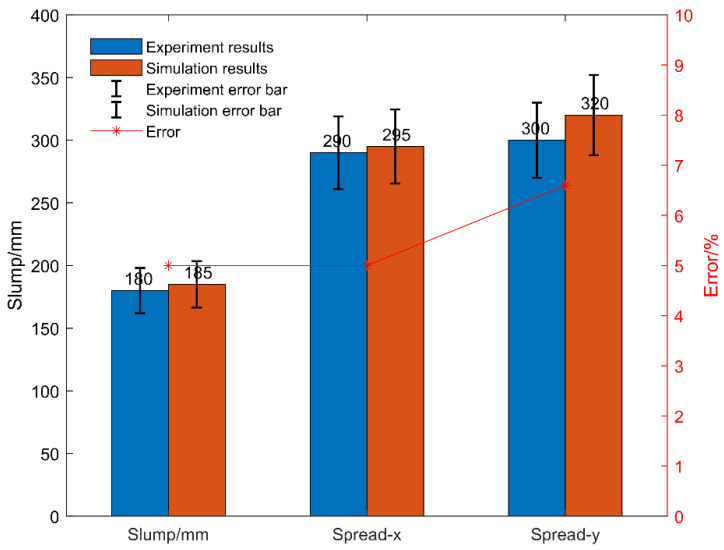
Error analysis of Group 2.

**Figure 15 materials-15-04294-f015:**
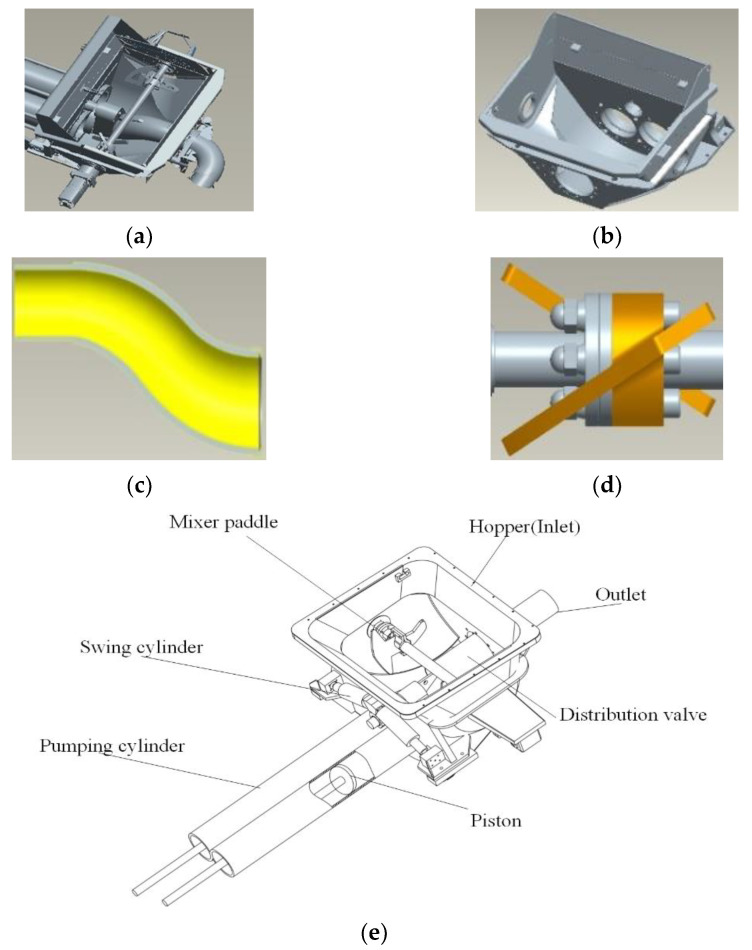
Concrete pumping machinery. (**a**) Concrete pumping assembly. (**b**) Hopper. (**c**) Distribution valve. (**d**) Mixer paddle. (**e**) Schematic of concrete machinery.

**Figure 16 materials-15-04294-f016:**
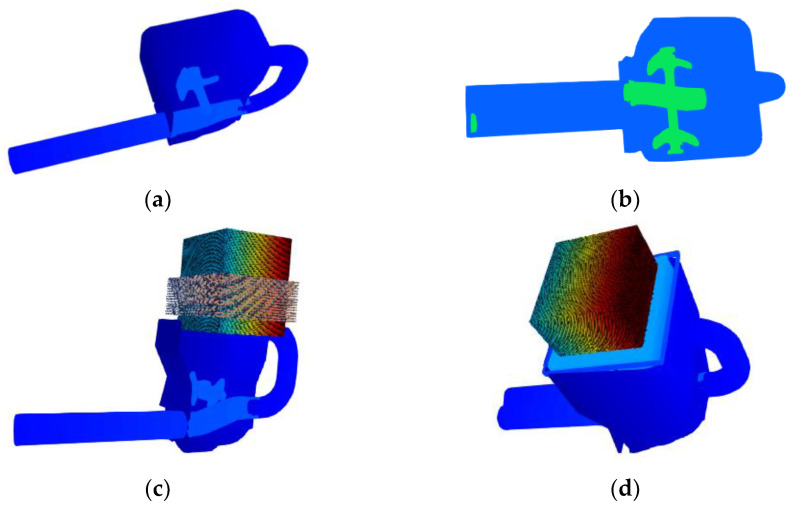
Simulation analysis model of concrete pumping. (**a**) Y–Y Cross-sectional view. (**b**) Complete machine. (**c**) SPH-DEM particles. (**d**) Whole model of concrete pumping.

**Figure 17 materials-15-04294-f017:**
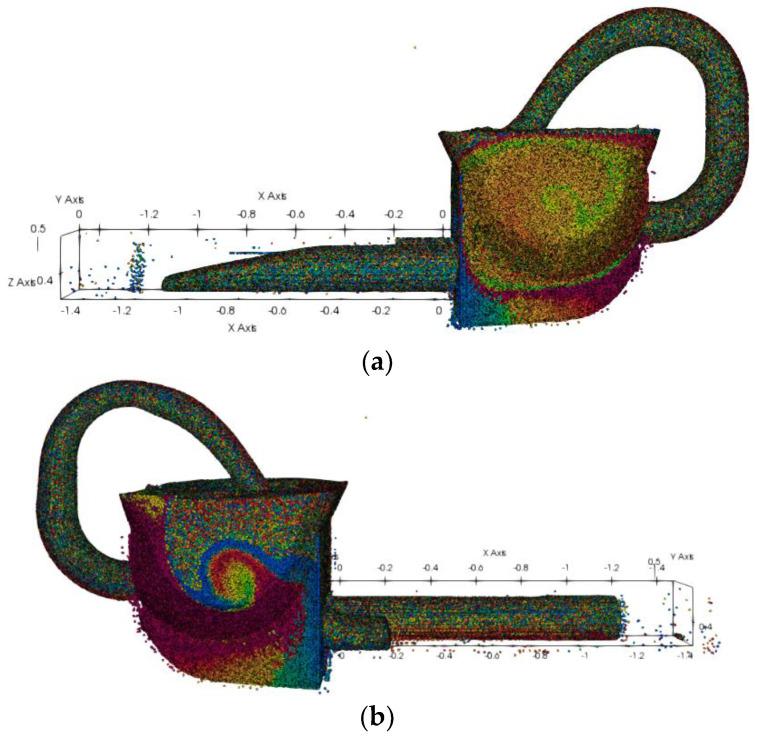
Concrete flow during pumping suction. (**a**) Flow state and velocity sampling; (**b**) Suction efficiency.

**Figure 18 materials-15-04294-f018:**
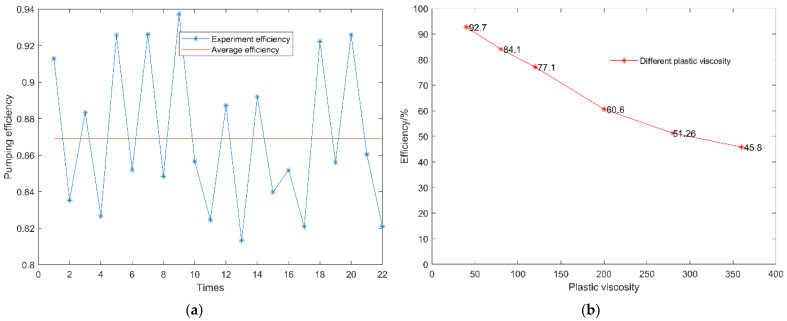
Experimental concrete pumping efficiency. (**a**) Simulation average efficiency. (**b**) Efficiency trends at different plastic viscosities.

**Figure 19 materials-15-04294-f019:**
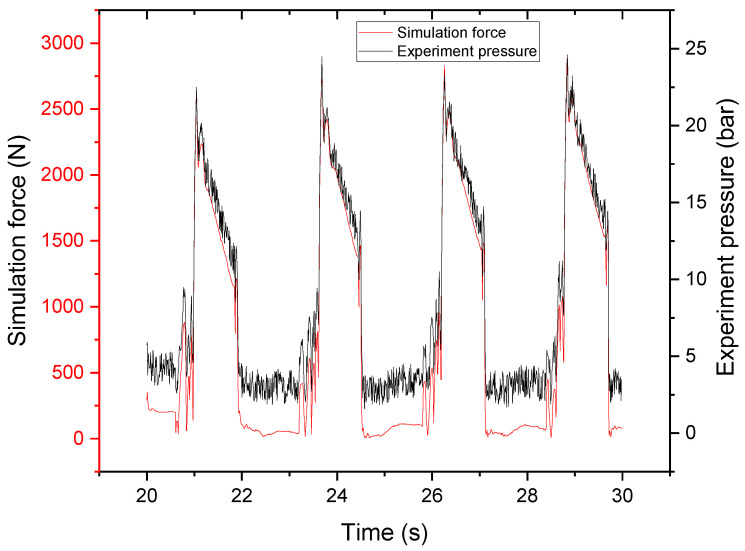
Pressure loss in experiment.

**Figure 20 materials-15-04294-f020:**
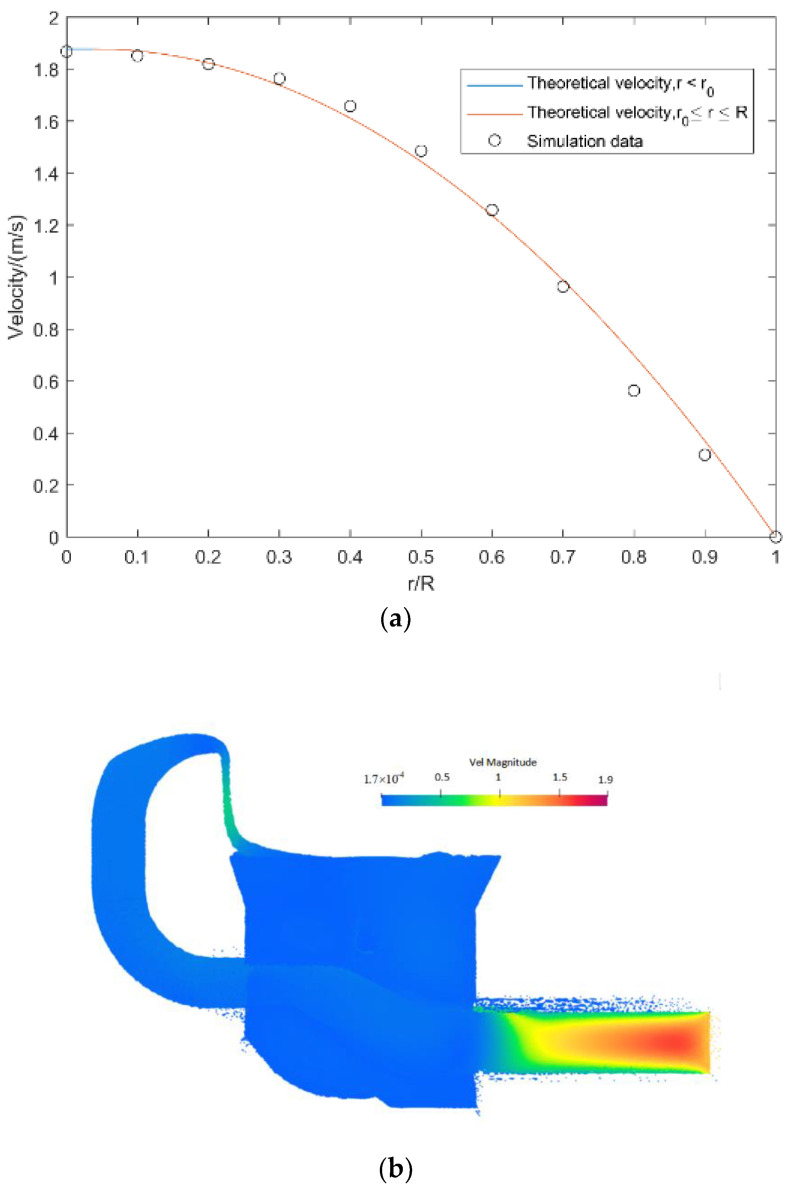
Simulated and analytical velocities of concrete pumping process. (**a**) Analytical laminar velocities. (**b**) Velocity cloud map.

**Table 1 materials-15-04294-t001:** Proportion of concrete (kg).

Groups	Water	Cement	Sand	Aggregate
1	3.6	9.6	16.7	20.1
2	4.5	7.1	21.7	17.5

**Table 2 materials-15-04294-t002:** Aggregate gradations.

Aggregate	19–16 mm	16–13.2 mm	13.2–9.5 mm	<9.5 mm
	31.3%	24.2%	21.2%	23.2%

**Table 3 materials-15-04294-t003:** Slump test results.

Number	Slump/(mm)	Dispersion/(mm)
1	190	350/375
2	180	290/295

**Table 4 materials-15-04294-t004:** Simulation parameters of SPH-DEM.

Parameters		Notation	Unit	Value
Simulation parameters	Number of fluid particles	$N_{pf}$		58,250
	Number of solid particles	$N_{ps}$		283,168
	Particle distance	$D_{p}$	m	0.004
	Smooth length	H	m	0.00692
	The ratio between smooth length and particle distance	K		1.73
	Simulation duration	t	s	10
	Constant of EOS	γ		7
	Sound speed coefficient	β		20
	The artificial viscosity coefficient	$\alpha_{II}$ $,\;\beta_{II}$		0.001
	Initial time interval	Δt	s	1 × 10^−6^
	CFL coefficient	$C_{FL}$		0.2
Rheological	Density	ρ	$\mathrm{kg}/m^{3}$	2040.0
parameters	Apparent dynamic viscosity	$\mu_{p}$	Pa·s	185
	Key coefficients of HBP model	m		100
		n		1
	Yield stress	$\tau_{y}$	Pa	282
DEM	Density	ρ	$\mathrm{kg}/m^{3}$	2300
Parameters	Young modulus	E	GPa	30
	Poisson rate	ν		0.3
	Restitution coefficient	r		0.1
	Kinetic friction coefficient	μ		0.4

## Data Availability

The data used to support the findings of this study are available from the corresponding author upon request.

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
