# Peer review of "Simulation Analysis of Concrete Pumping Based on Smooth Particle Hydrodynamics and Discrete Elements Method Coupling"

_materials, 2022, doi:10.3390/ma15124294_

Round 1
Reviewer 1 Report
Please see the attachment.

Author Response
Thank you very much for your positive and constructive comments and suggestions on our work.
Point 1:The manuscript should be improved grammatically.
Author response:This work has been polished by a native English speaker.
Point 2:Remove abbreviations from the title for better understanding of the audience.
Author response:Abbreviations from the title have been expansion, red font indicates reversion
Simulation Analysis of Concrete Pumping Based on Smooth Particle Hydrodynamics and Discrete Elements Method Coupling
Point 3:Add some quantitative results in the abstract.
Author response:We have revised the abstract in the manuscript as follows, red font indicates reversion.
Abstract: With the increase of suction efficiency of fresh concrete pumping in the confined space, the laminar flow state will be damaged by the return flow. This is a fact, but rarely studied. In this work, the flow state, flow velocity, and suction efficiency of the fresh concrete pumping are simulated using the coupled smooth particle hydrodynamics and Discrete Elements Method (SPH-DEM). The rheological parameters and (Herschel-Bulkley-Papanastasiou)HBP rheological model are adopted to simulate the fresh concrete in the numerical simulation model. The comparison reveals that the error between slump experimental result and that obtained by a numerical model under the HBP model is little. A model is established for numerical simulation of suction efficiency of fresh concrete pumping. An experimental concrete pumping platform is built, and the pressure and efficiency data during the pumping are collected. The comparison of numerical simulation with experimental results show that the error is less than 10%.
Point 4:add a paragraph related to research gap in the last paragraph of the introduction section and how your work is different from the existing once
Author response: Difference between with existing research has been added in front of last paragraph shown as red font.
In summary, previous studies of fresh concrete used DEM have given little attention on laminar flow state during the pumping. The conventional CFD method can’t correspond to the actual concrete composition ratios. On the other hand, most studies related to the CFD-DEM double-way coupling focused on discontinuing the state in pipes only. Therefore, finding new research means to describe the continued pumping and components of fresh concrete has been a recognized demand.
Point 5:Methodology and problem statement should be cleared
Author response:Statement of methods has been modified.
Point 6:Update the conclusions section in the view of conducted study
Author response: The conclusion has been rewritten and updated according to this work.
This work is aimed to determine a new method for fresh concrete behavior during the pumping.
1.The results show that the SPH-DEM method could be utilized to simulate fresh concrete through slump tests.The rheological parameters of fresh concrete are identified by a rheometer. The slump test error between simulation and experiment results is compared and is less than 10%. Therefore, SPH-DEM numerical simulation could respond to the real physical model.
A numerical simulation model of pumping process is established to analyze the effects of the changing in plastic viscosity of fresh concrete on the suction efficiency. In addition, the suction efficiency is studied experimentally. The numerical simulation results are compared with the experimental results, and the average suction efficiency error of suction efficiency is less than 5%.
The gradient variation of the pressure loss along the pipe (Dp/L) is calculated and the theoretical flow rate of concrete in the pipe is analyzed. Compared with the numerical simulation, the theoretical velocity analysis shows that the SPH-DEM numerical model is approaching the theoretical analysis. The issue of pipe blocking mechanism is an intriguing one which could be usefully explored in further research.
Thank you again for your valuable comments!

Reviewer 2 Report
1) The literature review should be revised and most recent similar works in the field should be introduced and their achievement should be presented.
2) At the end of Introduction, the available gaps should be presented clearly and then you should clearly address how you want to fulfill them. Finally, your novelties should clearly be presented.
3) In your methodology, you should present details of these item which are missed in your paper: How you solved governing equation (with software or developing code? And the other details)? What are the boundary conditions? The grid number indecency analysis should be added and the related results should be presented? The validation of the numerical simulation is missed (the results should be compared and validated by comparing with reliable works)?
4) Reliable references should be considered for your CFD governing equations. It is better to present the general form of the equations with more detail. Is your flow laminar or turbulent? In the case of turbulent, all of the details regarding turbulent model you used and the corresponding governing equations should be presented. These papers could help you to response these comments: https://doi.org/10.1016/j.molliq.2018.07.119 , https://doi.org/10.1016/j.jclepro.2018.07.127 , https://doi.org/10.1016/j.envpol.2018.07.027
Author Response
We sincerely thank you for your careful reading and comments.
point 1:The literature review should be revised and most recent similar works in the field should be introduced and their achievement should be presented.
Author response: For the pumping field at this stage are done with slurry and aggregate coupling in the pipe, literature such as Gram A[11] and Zhang Y[12]. But for the whole pumping there is almost no relevant work.
[11]Gram A, Silfwerbrand J. Numerical simulation of fresh SCC flow: applications[J]. Materials and Structures, 2011, 44(4): 805-813.
[12]Zhan Y, Gong J, Huang Y, et al. Numerical study on concrete pumping behavior via local flow simulation with discrete element method[J]. Materials, 2019, 12(9): 1415.
Point 2: At the end of Introduction, the available gaps should be presented clearly and then you should clearly address how you want to fulfill them. Finally, your novelties should clearly be presented.
Author response: Difference with existing research has been added in front of last paragraph shown as red font.
In summary, previous studies of fresh concrete used DEM have given little attention on laminar flow state during the pumping. The conventional CFD method can’t correspond to the actual concrete composition ratios. On the other hand, most studies related to the CFD-DEM double-way coupling focused on discontinuing the state in pipes only. Therefore, finding new research means to describe the continued pumping and components of fresh concrete has been a recognized demand.
Point 3: In your methodology, you should present details of these item which are missed in your paper: How you solved governing equation (with software or developing code? And the other details)? What are the boundary conditions? The grid number indecency analysis should be added and the related results should be presented? The validation of the numerical simulation is missed (the results should be compared and validated by comparing with reliable works)
Author response:The software adopted has been added in the last paragraph of introduction. The boundary condition adopted is DBC, which is described in the last paragraph in part 2.1. Details of the changes are shown in red font.
The suction efficiency of concrete pumping is numerically simulated in this work using coupled SPH-DEM (Based on opensource software DualSPHysics coupled with opensource software Project Chrono). Firstly, the SPH-DEM is utilized to create a fresh concrete simulation and analysis model. Secondly, practicality of the SPH-DEM to produce new concrete is proved by experimentally evaluating the numerical simulation features of the concrete using concrete aggregate grading, slump flow tests, and a rheometer. Then, the experiments are performed by using the verified completed concrete, and numerical models of the pumping process. Finally, the theoretical movement velocity of concrete movement in the cylinder is compared to that of the numerical simulation.
Point 4:Reliable references should be considered for your CFD governing equations. It is better to present the general form of the equations with more detail. Is your flow laminar or turbulent? In the case of turbulent, all of the details regarding turbulent model you used and the corresponding governing equations should be presented.
Author response:The specific details of the SPH solution to the N-S equation are in the literature presented by Monaghan J J [16] et al. Considering the limitation of the article length, it is not fully expanded here.
The flow of fresh concrete belongs to the category of laminar flow. Turbulent flow considerations are not concerned for this work. Velocity clouds has been added in Fig.20(b) to show the flow state of the laminar flow
[16] Monaghan J J. Smoothed particle hydrodynamics and its diverse applications[J]. Annual Review of Fluid Mechanics, 2012, 44: 323-346.
Thank you again for your valuable comments!

Reviewer 3 Report
the paper looked fine and very interesting. The analysis seemed rigourus and the results matched nicely to the experimental results. The only objection was the language. It was so hard for me to read, that I requested a deep languate revision. But with that, most probably the paper can be recommended for publication.
Author Response
We sincerely thank you for your reading and comments.
point 1:It was so hard for me to read, that I requested a deep languate revision.
Author response: This work has been polished by a native English speaker.
Thank you again for your precious comments!

Reviewer 4 Report
The manuscript is well-organized, just a few points:
- Polish the manuscript by a native-English speaker
- add at least 2 master tables to compare the results
- Add some schematics to show the mechanisms and trends
- Remove the old references
- Add the numbers with their standard deviations and RS
Author Response
We sincerely thank you for your careful reading and comments.
Point 1:Polish the manuscript by a native-English speaker
Author response:This work has been polished by a native English speaker.
Point 2:add at least 2 master tables to compare the results.
Author response: Fig.18(b) have been added to visualization table to compared suction efficiency at different plastic viscosities.
Fig.20(b) has been added to visualization table to show the results compared with analytical larminar velocities.
Point 3:Add some schematics to show the mechanisms and trends
Author response: Fig.8(b) has been added to show the hydraulic system schematics of concrete pumping platform.
Fig18.(b) has been added to show the efficiency trends at different plastic viscosities.
Point 4:Remove the old references
Author response: Old references has been removed.
point 5:Add the numbers with their standard deviations and RS
Author response: Number has been added in Fig.10, Fig.12, Fig.14 and Fig.18(b). Deviations and RS has been added in Fig12 and Fig14.
Thank you again for your precious comments!

Round 2
Reviewer 2 Report
The authors failed to address my comments and I can't suggest acceptance of this paper.
Author Response
Please see the author coverletter.

Reviewer 3 Report
This paper proposes a computational method to model the pumping process of concrete. The computational algorithm is based on the combination of the SPH and DEM methods, together with a rheological model for concrete. Furthermore, the work compares the computational results with specially developed experimental tests. The accuracy of the computational results is impressive, which makes this research work very valuable to the scientific community. Thus, the high value of the paper should warrant publication.
However, before publication, I would ask to the authors to address these issues.
- Although the clarity of the paper has dramatically improved, a second revision of the writting would be very beneficial and will increase the impact of this work. Bellow, I include some suggestions.
- In general, the figure legends and labels are too small. In particular, see Figs. 2, 3, 7, 8b, 18, 19, 20b.
- Abstract, L. 12. Here, return flow is mentioned, but for a reader out of the field, it is hard to understand what it means. L. 17, I would change "The comparison" by "The study".
- P. 1, L 28. I would just say "improve performance"
- P. 2 L 12. I wouls substitute "discretes" by "discretizes"
- P. 2 L 14. There is an extra dot between [10] and mixed.
- P 2, L 30. instanton ?
-P. 2, L 39-40 are barely
- P. 3 L 9, "simulaton"
- P. 3 L 14-15 are not clear.
- P. 3, L 20-21. I would remove "based on its appearance and relationship"
- P 12 L 9. Should Fig 13 be Fig 14?
- P 12 L 17. ssimulation ?
- P 13, Fig 15. This set of figures is not enough for me to understand the geometry of the pump. Furthermore where do (c) and (d) go? Where is the inlet and outlet?
How much is modeled of all this? How about the interior of the reciprocating pump?
- P 13 L 21. Should Fig 15 be Fig 16?
- Conclusion L 7-8. I would write "during pumping"
- P 16 L 15. Do you refer with "changing" to "variation" ?
Reviewer 4 Report
Accept
Author Response
Thanks for your acceptance!